# Using Physiologically Based Pharmacokinetic Models for Assessing Pharmacokinetic Drug–Drug Interactions in Patients with Chronic Heart Failure Taking Narrow Therapeutic Window Drugs

**DOI:** 10.3390/ph18040477

**Published:** 2025-03-27

**Authors:** Nadezhda Hvarchanova, Maya Radeva-Ilieva, Kaloyan D. Georgiev

**Affiliations:** Department of Pharmacology, Toxicology and Pharmacotherapy, Faculty of Pharmacy, Medical University—Varna, 9000 Varna, Bulgaria; nadejda.hvarchanova@mu-varna.bg (N.H.); maya.radeva@mu-varna.bg (M.R.-I.)

**Keywords:** physiologically based pharmacokinetic (PBPK) modeling, potential drug–drug interactions (pDDIs), potential pharmacokinetic drug–drug interactions (pPKDDIs), CYP3A4, CYP2C9, P-gp

## Abstract

**Background:** Pharmacotherapy of chronic heart failure (CHF) with a reduced ejection fraction includes a combination of drugs. Often, different groups of drugs are added together for the treatment of concomitant conditions, such as statins, anticoagulants, antiplatelet agents, and cardiac glycosides, which have a narrow therapeutic window. Increased medication intake is a prerequisite for the increased risk of potential adverse drug–drug interactions (DDI), especially those occurring at the pharmacokinetic level. The main objectives of this study are to identify the most common potential pharmacokinetic drug–drug interactions (pPKDDIs), to explore more complex cases, and to simulate and analyze them with appropriate software. **Methods:** The data selected for the simulations were collected over a two-year period from January 2014 to December 2015. Identification of the pPKDDIs was carried out using Lexicomp Drug interaction, while simulations were performed with Simcyp software (V20, R1). **Results:** The most common pharmacokinetic mechanisms responsible for the occurrence of drug–drug interactions in the selected drugs with narrow therapeutic windows are those featuring the cytochrome isoforms CYP3A4 and 2C9 and the efflux pump—P-glycoprotein (P-gp). Simulations with the available data in Simcyp software showed possibilities to analyze and evaluate pPKDDIs, which would be difficult to assess without appropriate software, as well as ways to manage them. **Conclusions:** The frequency and complexity of pPKDDIs in patients with cardiovascular disease are high. Therefore, such patients require a specific approach to reduce these risks as well as to optimize the therapy. An appropriate PBPK software with the necessary database would be suitable in these cases.

## 1. Introduction

The frequency of patients suffering from heart failure (HF) is constantly growing, and the disease is currently affecting more than 65 million people worldwide [1]. Increasing incidence occurs due to the aging of the population, improved survival, and the treatment of heart diseases, such as coronary heart disease, hypertension, and valvular heart disease, and, as a consequence, the prolonged life expectancy of patients with HF. Despite advances in management, the five-year survival rate for these patients is about 50% and is comparable to that of the most common neoplastic diseases in men and women [2].

According to the new guidelines for the treatment of acute and chronic heart failure of the European Society of Cardiology (ESC), pharmacotherapy of the most common form of heart failure, with reduced ejection fraction (HFrEF), has been significantly changed compared to the previous version, and now includes the prescription of four pharmacological classes of drugs immediately after the diagnosis [3]. The four classes of basic therapy that have been shown to improve survival, decrease sudden death, and prevent heart failure hospitalizations include: (1) angiotensin-converting enzyme inhibitor (ACE-I) or angiotensin receptor/neprilysin inhibitor (ARNI) such as valsartan/sacubitril; (2) beta-blockers; (3) mineralocorticoid receptor antagonists (MRA); (4) sodium-glucose co-transporter 2 inhibitors (SGLT2-inhibitors). In addition to the standard treatment, there is often a need to prescribe additional drugs to treat patients’ concomitant conditions, such as statins, anticoagulants, antithrombotics, cardiac glycosides, etc. [3,4]. These drugs, despite their proven effectiveness, belong to the group of high-risk medications with a narrow therapeutic window, which makes them vulnerable to drug–drug interactions. Therefore, it is clear that patients with chronic heart failure will receive polypharmacy to control the disease, and they should be very closely monitored for increased risk of drug–drug interactions, especially when high-risk narrow therapeutic window drugs are included in the therapy [5,6].

Physiologically based pharmacokinetic (PBPK) modeling (in silico) is now an integral part of the drug discovery and development process, and is widely used to quantitatively predict drug–drug interactions from in vitro (“bottom-up”) and in vivo clinical experiments (“top-down”) [7,8,9]. Thanks to these models, many drugs have been approved by global regulators in lieu of clinical studies. Despite the widespread use of the PBPK model in regulatory practices, its use in clinical settings is generally hampered by the difficulty of use of such complex software by most clinicians [10]. However, using such software, especially in problem heart failure patients, such as those receiving high-risk narrow therapeutic window drugs, would justify the need for a specialist, such as a hospital/clinical pharmacist who would be trained in this area and could be valuable in the multidisciplinary team providing the pharmaceutical care [11,12].

The aim of the study is to assess the risk of potential pharmacokinetic drug–drug interactions (pPKDDIs) in chronic heart failure (HF) patients and the performance of computational models such as physiologically based pharmacokinetic (PBPK) modeling to evaluate them. For this purpose, data from patients receiving drugs with a narrow therapeutic window was used and PBPK models were developed and validated.

## 2. Results

### 2.1. Evaluation of Drugs with a Narrow Therapeutic Window for pDDIs in Patients with CHF

Table 1 presents the most common potential drug–drug interactions (pDDIs) in selected patients associated with the coadministration of drugs with a narrow therapeutic range. The putative mechanisms of the interaction are included. The results are obtained by Lexicomp software.

As can be seen from Table 1, the most common pharmacokinetic mechanisms responsible for drug interactions in the selected patients, taking narrow therapeutic window drugs, are those related to the cytochrome P450 isoforms CYP3A4 and 2C9 and the efflux transporter or pump, multidrug resistance protein 1 (MDR1), also known as P-gp (permeability glycoprotein).

### 2.2. Simulation of CYP3A4 Interactions with Simcyp^®^ Software

The three statins that are metabolized by CYP3A4 are simvastatin, lovastatin, and atorvastatin. Their peculiarities are that simvastatin and lovastatin undergo intensive first-pass metabolism (bioavailability < 5%), while atorvastatin is less pronounced (bioavailability ~12–14%) [13]; because of this, simvastatin and lovastatin are more vulnerable to drug–drug interactions with CYP3A4 inhibitors. The Simcyp^®^ software was provided with simvastatin data only, so the simulations presented below apply to it (Table 2).

All three mentioned statins are contraindicated (risk category X, severe reaction according to the Lexicomp software) with potent CYP3A4 inhibitors, such as clarithromycin, ketoconazole, itraconazole, ritonavir, atazanavir, etc., due to the risk of muscle toxicity (including rhabdomyolysis). In the simulations performed, the main reason for the increased simvastatin exposure is due to the inhibition of its first-pass metabolism and the increased bioavailability (e.g., Fg × Fh is approximately 10× increased in the simulation with ketoconazole) rather than inhibition of systemic clearance (CL) (~2× inflated in the same case).

Simulations were also conducted using simvastatin alongside the concurrent co-administration of various CYP3A4 inhibitors. The results are presented in Figure 1.

What is observed from Figure 1 is that the addition of an inhibitor of the CYP3A4 enzyme leads to a summation of the activities of the individual inhibitors with respect to CYP3A4, as expressed by the CmaxR and AUCR values, and redirects the elimination of simvastatin to an alternative pathway (in this case by additional liver microsomes). In this case, the alteration in simvastatin metabolism was unable to offset its elimination, leading to substantial increases in pharmacokinetic parameters (Cmax and AUC) when inhibitors were used together.

### 2.3. Simulation of CYP2C9 Interactions with Simcyp^®^ Software

The major route of elimination of coumarin anticoagulants (warfarin, acenocoumarol) is through metabolism by CYP2C9, with a small fraction eliminated unchanged in the urine. In Bulgaria, the main used drug is acenocoumarol. Peculiarities in the pharmacokinetics of both drugs are shown in Table 3.

Both drugs have one chiral center, which determines the presence of two enantiomers, of which the S-enantiomer is about 2 to 5 times more potent [14]. The provided software contains data on the physicochemical and pharmacokinetic properties of S-warfarin, and the simulations align with them. CYP2C9 has multiple allelic variants, the most common being CYP2C9*2 and CYP2C9*3. Both variants show reduced enzyme activity and occur at a high frequency in Caucasian populations compared to African American and Asian populations [15]. Table 4 presents the data on pharmacogenetic differences in CYP2C9 available in the Simcyp^®^ software for individual populations, while Table 5 shows the simulations performed with 10 mg S-warfarin.

As can be seen from both tables, in the Caucasian race, the ratio of the phenotype of extensive to poor metabolizers is 0.94 to 0.06, while in the Asian race, the ratio is 0.99 to 0.01. However, in the latter, Cmax and AUC are slightly higher. The differences come from the amount of enzymes (CYP2C9) that are present in the liver. In the European Caucasian race, they are 73 pmol/mg protein for intensive and 29 pmol/mg protein for poor metabolizers, while in the Asian race, the values are 60 pmol/mg protein and 24 pmol/mg protein, respectively, hence the reduced intensity of metabolism and the different CL values of S-warfarin in the individual populations (CL_NEurC_ = 0.24 ± 0.22 L/h vs. CL_Chinese_ = 0.17 ± 0.20 L/h) [16]. The demonstration with CYP2C9 inhibitors reinforces this and shows a significantly greater risk of drug–drug interactions in European Caucasians compared to Asians if equal doses of warfarin are used (Table 6).

### 2.4. Simulation of P-gp Interactions with Simcyp^®^ Software

Digoxin and dabigatran etexilate are substrates of the efflux pump P-gp (MDR1/ABCB1) and are at risk of drug–drug interactions with P-gp inhibitors. Interactions of digoxin and dabigatran etexilate with verapamil (a P-gp inhibitor) were simulated. The Simcyp^®^ simulator contains complete data on verapamil and its main metabolite, norverapamil, which is a 3–4 times more potent inhibitor of P-gp [17]. Lexicomp places the interaction of verapamil with digoxin in category C, while that with dabigatran etexilate in category D (one interaction was detected for 2014; Table 1).

Several simulations were performed with digoxin 0.5 mg single dose alone and in combination with verapamil/norverapamil 240 mg daily (80 mg/8 h), with the deactivation of the inhibition either in the liver or in the gastrointestinal tract (GIT) in order to follow the importance of P-gp localization on the development of the interaction (Table 7). The results presented in Table 7 show that verapamil, despite being less potent, interacts with digoxin as strongly as its metabolite norverapamil. The interaction was mainly due to the inhibition of P-gp in the GIT and not so much in the liver, which was tracked by sequentially excluding inhibition in the GIT and liver. Another way to confirm this is by changing the application path. If the same dose of verapamil is simulated for i.v. bolus administration (of course, the intravenous therapeutic dose of verapamil is significantly smaller, but for the purpose of demonstrating the thesis the simulation was undertaken with this dose), the interaction is less pronounced.

Two cases were simulated with the Simcyp^®^ software— the administration of 150 mg of dabigatran etexilate with a daily 240 mg dose of verapamil/norverapamil (80 mg/8 h) at the same time and 2 h after. Dabigatran etexilate is a prodrug that is converted due to the action of carboxyesterases into an active metabolite, dabigatran. The Simcyp software also had the necessary physicochemical and pharmacokinetic data for this active metabolite. The results are summarized in Table 8.

As shown, the simultaneous administration of verapamil with dabigatran etexilate leads to a significant increase in the potential for the interaction between the two drugs, expressed by the AUCR and CmaxR values of the prodrug dabigatran etexilate and the active metabolite dabigatran. Conversely, the administration of verapamil 2 h after dabigatran etexilate reduces the risk of such interaction.

## 3. Discussion

Pharmacokinetic drug–drug interactions are difficult to identify, predict, and manage in clinical practice, especially when it comes to complex ones, involving metabolizing enzymes and/or transport systems, and when the drugs used have a narrow therapeutic window [18,19]. Available software programs for drug–drug interaction analysis generally provide a rough assessment of these interactions as either major, moderate, minor, or no interaction risk without taking into account disease specificity, sex, age, genetic characteristics of the patient population, etc., as well as the dose, dosage regimen, route of administration, etc., of the used drugs. Because of this, in more complex cases, there is a need for a software program that can simulate the interactions, and, based on the received information, the attending physician can modify the dosage regimen and optimize the therapy.

Software capable of generating physiologically based pharmacokinetic (PBPK) models of drugs and simulating pharmacokinetic drug–drug interactions could be valuable in these cases. The wealth of data integrated into the Simcyp^®^ software allows it to simulate multiple pharmacokinetic drug–drug interactions related to metabolizing enzymes and/or transport proteins, which makes it very suitable in clinical practice to assess possible drug–drug interactions in high-risk patients. The software’s data reflect changes in key pharmacokinetic parameters, such as Cmax and AUC, which are essential for assessing pharmacokinetic interactions. It also incorporates factors influencing these parameters, including absorption rate, first-pass metabolism, solubility, permeability, dosage form, food effects, and stomach pH. In the presented examples, some of the capabilities of the software were demonstrated, such as (1) consideration of the opportunities to include alternative pathways of elimination when inhibiting the drug’s primary pathway of elimination; (2) determining the drugs that are the most sensitive and the most resistant to interactions in a given population, depending on the gene and phenotypic manifestations; (3) the possibility of simulating the drug–drug interactions of the main compounds and their active metabolites; (4) the possibility to add several inhibitors simultaneously in the simulations (when there is a similar clinical situation for example); (5) the possibility of changing the route of administration, or the dosage form used, as well as changing the timing of the appointment of doses during the day to see how the responses change.

Five cases of rhabdomyolysis have been reported in patients taking simvastatin, that developed between day 7 and week 4 after starting ketoconazole [20]. Cases of rhabdomyolysis have also been described in patients taking simvastatin and itraconazole [21]. In a two-phase crossover study of 10 healthy subjects who received 200 mg of itraconazole daily or placebo for 4 days, with a single dose of 40 mg simvastatin on day 4, peak serum levels of total simvastatin acid (simvastatin acid + simvastatin lactone) increased 17-fold, and AUC increased 19-fold (*p* < 0.001) [22]. In simulations performed with Simcyp^®^, Cmax, and AUC of simvastatin co-administered with itraconazole they were approximately 21× increased, respectively. The protease inhibitors, atazanavir, and ritonavir, have not been tested for drug interactions with simvastatin in clinical studies with healthy volunteers. In most clinical studies, there was concomitant administration of other medications or ritonavir, which was used with another protease inhibitor as a booster; therefore, the cause of the interaction cannot be inferred [23]. In a study of 31 healthy volunteers who received either 10 mg of atorvastatin daily or 20 mg of simvastatin daily for 28 days, with 1250 mg of nelfinavir twice daily during the last 14 days, the AUC of atorvastatin was approximately twofold increased, and the Cmax and the AUC of simvastatin approximately six-fold [24]. In the simulations performed with Simcyp^®^ software, the AUC of simvastatin was approximately six-fold higher than in the clinical studies. The simultaneous use of macrolide antibiotics (erythromycin or clarithromycin) with statins (simvastatin, lovastatin, or atorvastatin) leads to an increased risk of hospitalizations due to rhabdomyolysis, acute renal failure, or an increased incidence of a fatal outcome [25]. In clinical studies with healthy volunteers, the Cmax and AUC of simvastatin increased approximately 2.3- and 3.9-fold, respectively, when used simultaneously with clarithromycin [26]. In the virtual simulations performed, the PBPK model was used with a single simultaneous administration of both drugs and showed a 2.2- and 2.6-fold rise in simvastatin Cmax and AUC values. Conducting such simulations in clinical practice would help to illustrate the interaction better so that the physician can appropriately adjust the dosage regimens of the drugs used.

Drugs that are primarily metabolized by cytochrome CYP2C9 isoform and have a narrow therapeutic window are coumarin anticoagulants (acenocoumarol, warfarin), sulfonylureas (such as gliclazide, etc.), and phenytoin. When a patient has one of the two polymorphic forms (CYP2C9*2 or CYP2C9*3), the doses of these drugs need to be lowered because drug clearance is reduced, and plasma concentrations will be increased. Also, patients with these polymorphic forms may have an inadequate drug response when taking prodrugs that require CYP2C9-mediated bioactivation to exert a therapeutic effect (such as losartan and cyclophosphamide) [27]. The advantage of the Simcyp^®^ software is that it has the ability to simulate pharmacokinetic behavior, taking into account precisely these pharmacogenetic differences in individual enzyme systems between the different races based on accumulated epidemiological data. It takes into account the amount of enzymes (CYP2C9) that are present in the liver as well. In clinical studies with healthy volunteers, the interaction of warfarin and fluconazole has been demonstrated. Fluconazole administered in doses of 100, 200, and 400 mg showed an increase in the AUC of S-warfarin by 35%, 86%, and 100%, respectively [28]. By performing simulations of fluconazole and S-warfarin in European Caucasian and Chinese populations, it was found that the risk of drug interactions is greater in European Caucatioans than in Asians due to lower plasma concentrations (Table 4 and Table 5). When reviewing the results in detail, those most sensitive to the interaction (those with the highest AUCR values > 3) were logically those with the CYP2C9*1/*1 genotype (rapid metabolizers) and with the highest amount of CYP2C9 enzymes (16,890,713.14–17,832,696.92 pmol). The lowest degree of interaction (those with the lowest AUCR values) was observed in patients with the CYP2C9*3/*3 genotype (poor metabolizers) with AUCR values in the range of 1.00–1.02 and those with the CYP2C9* genotype 1/*1 with the lowest enzyme expression (566,035.71–913,298.10 pmol), with AUCR values of 1.05–1.07. Performing the simulation in such cases would be useful not only in assessing the drug–drug interactions themselves, but also in evaluating the therapeutic response or the observed adverse effects.

In clinical studies with healthy volunteers, when using daily doses of 240 mg of verapamil, digoxin levels increased by 60–80% [29]. Simulations performed with a single dose of 0.5 mg digoxin alone and in combination with 240 mg of verapamil/norverapamil daily (80 mg/8 h) showed similar results to those reported in the clinical trials (Table 7). In order to trace the involvement of the parent compound (verapamil) and its metabolite (norverapamil) and the level of interaction (inhibition of P-gp in the GIT or in the liver), simulations were performed with the successive exclusion of possible P-gp inhibitions. The results show that verapamil, despite being less potent, interacts with digoxin as strongly as its metabolite norverapamil. The interaction occurred mainly due to the inhibition of P-gp in the GIT and not so much in the liver, which was tracked by sequentially excluding inhibition in the GIT and the liver. Another way to confirm this is by changing the application path (as demonstrated in Table 7). In healthy volunteers, the interaction of dabigatran etexilate at a dose of 150 mg once daily together with verapamil was also studied in different doses, with different durations, in different dosage forms, and at different times. The smallest interaction potential was determined when dabigatran etexilate was administered 2 h before verapamil, and, accordingly, the greatest was when verapamil was administered one hour before dabigatran etexilate [30]. Two cases were simulated with the Simcyp^®^ software— the administration of 150 mg of dabigatran etexilate with a 240 mg daily dose of verapamil/norverapamil (80 mg/8 h) at the same time and 2 h after. Dabigatran etexilate is a prodrug that is converted under the action of carboxyesterases into an active metabolite—dabigatran. The Simcyp^®^ software also had the necessary physicochemical and pharmacokinetic data for this active metabolite. As shown by the results of the simulations, the simultaneous administration of verapamil with dabigatran etexilate leads to a significant increase in the potential for interaction between the two drugs, expressed by the AUCR and CmaxR values of the prodrug dabigatran etexilate and the active metabolite dabigatran in Table 8. Conversely, the administration of verapamil 2 h after dabigatran etexilate reduces the risk of interaction, which is also confirmed by the above-cited clinical trial [30]. Carrying out these simulations in a clinical practice would provide a better demonstration of the interaction, enabling the physician to make more informed adjustments to appropriately modify the drug dosage regimens in order to avoid possible interaction.

## 4. Materials and Methods

### 4.1. Data and Software

The data used have already been described separately in our previous publications [31,32,33], while in this one, they are united. The medical records from the Cardiology Clinic at St. Marina University Hospital from a two-year period (January 2014–December 2015) were retrospectively examined for potential drug–drug interactions. In total, the discharge summaries of 1956 patients were reviewed (985 for 2014 and 971 for 2015). Patient selection was based on the following criteria: (1) diagnosis of HF in NYHA (New York Heart Association) class 2–4; (2) receiving standard treatment (e.g., ACE-inhibitors or AT1-blockers and/or beta-blockers and/or diuretics and/or aldosterone antagonists and/or nitrates and/or ivabradine); (3) simultaneous use of medications with a risk of potential drug–drug interactions due to complex pharmacological properties and life-threatening adverse effects, such as statins, anticoagulants (acenocoumarol or new oral anticoagulants, NOACs), and antithrombotic drugs (clopidogrel or ticagrelor) (Figure 2).

The Committee on Research Ethics at the Medical University “Paraskev Stoyanov” in Varna, Bulgaria, has approved the use of the data in this study. The complete patient information was kept private and not available to the public.

Based on the set criteria, a total of 487 patients were selected for the study (239 for 2014 and 248 for 2015). Identification of pDDIs was performed using Lexicomp^®^ Drug Interactions (Wolters Kluwer, Hudson, OH, USA) software. The software organized the interactions into five groups: A (No known interactions), B (No action required), C (Monitor therapy), D (Consider modifying therapy), and X (Avoid combination). Additionally, each interaction was rated by the severity of the reaction, which was categorized into four levels: major, moderate, minor, and N/A. The interaction descriptions also included a paragraph outlining the presumed mechanism behind each interaction.

After the analysis, the most common pharmacokinetic mechanisms responsible for drug–drug interactions in the selected drugs with a narrow therapeutic window were determined, and they included the cytochrome isoforms CYP3A4 and 2C9, as well as the efflux pump—P-gp.

Simcyp^®^ software (Version 20, Release 1) was used to conduct some of the potential pharmacokinetic drug–drug interactions (pPKDDIs) that could be simulated with the available data in the provided version of the software. The working principle of the Simcyp^®^ is described in detail elsewhere [34] or on the commercially available site [35]. In short, the model’s design was pre-integrated into a simulator that, by using physicochemical parameters, in vitro and in vivo data, and/or in silico predictors, allows the evaluation of the drug absorption, distribution, metabolism, and excretion (ADME) processes. Unless stated otherwise, the conditions and assumptions applied in the studies were primarily based on the worst-case scenarios (when the two interacting drugs reached peak concentrations). For the purpose of the study, PBPK models were developed and validated. All simulations, unless otherwise noted, were performed on a virtual population of healthy volunteers available in Simcyp^®^ without modifications, aged 50 to 65 years, with 50% being female. The number of studies is 10, with 10 subjects each, or a total of 100 studies conducted. The main outcome measure is the evaluation of pPKDDIs.

### 4.2. Data Analysis

The two main pharmacokinetic parameters commonly used as the basis for the prediction of pPKDDIs are the area under the curve (AUC) and the maximum plasma concentration (Cmax). The simulator features automatic generation of mean values of the ratios (AUCR and CmaxR) with a confidence interval in the absence and presence of inhibitor (e.g., AUCi/AUC, where AUCi and AUC are the AUC [0–∞] values of the substrate in the presence and absence of inhibitor, respectively).

## 5. Limitations and Future Research

In our opinion, the main strength of this study is the demonstration of the opportunity to implement appropriate software that benefits physicians in complex situations, whenever there is a need to simulate a clinical situation (in the case of pharmacokinetic drug–drug interactions but not limited to this) to adequately prescribe the appropriate drug. At the same time, it takes into account the multiple features that may lead to drug–drug interactions, such as dose, route, and time of application of the drug, as well as race of the patient and pharmacogenetic features. As a basis, patients with heart failure were used. In these patients, the disease predisposes to an increased risk of drug–drug interactions, and the application of such software will improve the overall therapy. However, our study has certain limitations. One of them is that the simulations performed only use the available data provided in the software, without changing its settings or adding new data. The patient data included in the study were retrospectively collected, and they were not monitored over time, so there are no data after discharge from the hospital.

Our future plans include achieving close interprofessional collaboration with cardiology physicians and, in particular, the ones treating patients with heart failure in cardiology clinics. This will enable the prospective analysis of the prescribed therapies for potential drug–drug interactions in high-risk patients while showcasing the capabilities of the PBPK software.

## 6. Conclusions

The frequency of drug–drug interactions in patients with HF is high. The therapy of these patients should be carefully monitored, verified, and controlled for possible drug–drug interactions with the aid of appropriate software. In complex cases involving high-risk patients and those receiving high-risk drugs, PBPK software such as Simcyp would be suitable for more precise measurements of pharmacokinetic parameters (Cmax and AUC), which can lead to better-adjusted dosing regimens for the interacting drugs. Considering the known complexity of handling such software, training a suitable specialist, such as a hospital or clinical pharmacist, would facilitate its adequate use.

## Figures and Tables

**Figure 1 pharmaceuticals-18-00477-f001:**
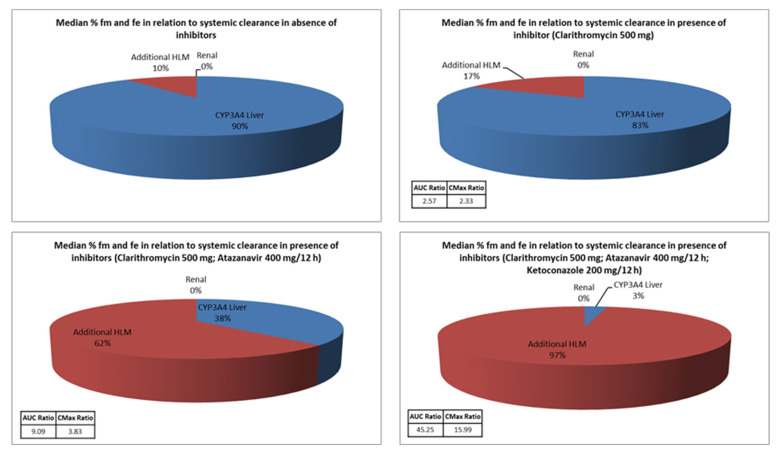
Simulations of simvastatin with the simultaneous administration of several CYP3A4 inhibitors. HLM—human liver microsomes, fm—fraction which is subjected to metabolism, fe—fraction that is excreted unchanged through the kidneys.

**Figure 2 pharmaceuticals-18-00477-f002:**
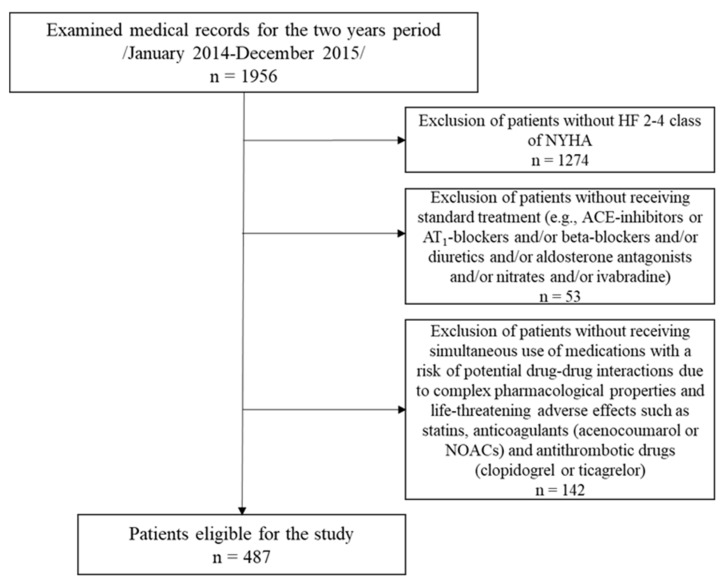
Flow chart of the patients selected for the study.

**Table 1 pharmaceuticals-18-00477-t001:** Potential drug–drug interactions in HF patients (detected by Lexicomp) associated with co-prescribed narrow therapeutic window drugs. The calculated percentages are based on the selected patients in the study.

Potential Drug–Drug Interactions	Severity/Risk Category	Frequency for 2014	Frequency for 2015	Mechanism of Interaction
**HMG-CoA reductase inhibitors (statins)**				
Simvastatin + 1,4-DHP-CCB	Major/D	14 (5.9%)	6 (2.4%)	CYP3A4
Statin + Colchicine	Major/D	1 (0.4%)	-	CYP3A4/OAT/PD
Statin + Fenofibrate	Major/C	12 (5%)	9 (3.6%)	PD
Rosuvastin + Amiodarone	Major/B	-	3 (1.2%)	CYP2C9
Statin + Verapamil	Major/D	-	3 (1.2%)	CYP3A4
**Anticoagulants (coumarins)**				
Acenocoumarol + Allopurinol	Moderate/D	4 (1.7%)	2 (0.8%)	CYP2C9
Acenocoumarol + SMZ/TMP	Major/D	1 (0.4%)	-	CYP2C9, PPB/PD
Acenocoumarol + Fenofibrate	Major/D	1 (0.4%)	4 (1.6%)	CYP2C9
Acenocoumarol + Amiodarone	Major/D	2 (0.8%)	4 (1.6%)	CYP2C9
Acenocoumarol + Thyreostatic	Moderate/D	2 (0.8%)	2 (0.8%)	PD
Acenocoumarol + NSAIDs	Moderate/D	-	3 (1.2%)	N/A
**New oral anticoagulants (NOACs)**				
Apixaban + Aspirin	Major/D	-	2 (0.8%)	PD
Apixaban + Clopidogrel	Major/D	-	2 (0.8%)	PD?
Dabigatran + Amiodarone	Major/D	1 (0.4%)	2 (0.8%)	P-gp
Dabigatran + Verapamil	Major/D	1 (0.4%)	-	P-gp
Dabigatran + Carvedilol	Major/D	1 (0.4%)	4 (1.6%)	P-gp
Dabigatran + Aspirin	Major/D	-	2 (0.8%)	PD
Dabigatran + Fluconazole	Major/C	-	1 (0.4%)	CYP3A4?, P-gp?
Rivaroxaban + Verapamil	Major/D	1 (0.4%)	-	CYP3A4, P-gp
**Antithrombotic drugs**				
Clopidogrel + PPI	Moderate/D	17 (7.1%)	24 (9.7%)	CYP2C19
Clopidogrel + Aspirin	Moderate/C	12 (5%)	10 (4%)	PD?
Ticagrelor + Aspirin	Major/D	-	1 (0.4%)	PD?
**Cardiac glycosides**				
Digoxin + Amiodarone	Major/D	1 (0.4%)	1 (0.4%)	P-gp
Digoxin + Telmisartan	Moderate/C	4 (1.7%)	3 (1.2%)	P-gp
Digoxin + Colchicine	Moderate/C	1 (0.4%)	-	P-gp

**Table 2 pharmaceuticals-18-00477-t002:** Simulations of the simultaneous use of simvastatin with classic CYP3A4 inhibitors. The values are presented as ratios—AUCR (AUCi/AUC) and CmaxR (Cmax,i/Cmax)—in the presence and absence of inhibitors.

Inhibitors of CYP3A4	AUCR ± SD	CmaxR ± SD
Clarithromycin 500 mg/24 h	2.57 ± 0.51	2.33 ± 0.41
Ketoconazole 200 mg/12 h	28.42 ± 11.85	14.02 ± 5.90
Ketoconazole 400 mg/24 h	34.10 ± 15.75	15.77 ± 7.08
Itraconazole 200 mg/24 h	20.63 ± 8.79	11.56 ± 4.47
Atazanavir 400 mg/12 h	5.49 ± 3.75	2.28 ± 1.29
Ritonavir 100 mg/12 h	5.85 ± 3.59	2.67 ± 1.13

**Table 3 pharmaceuticals-18-00477-t003:** Physicochemical and pharmacokinetic parameters of the vitamin K antagonist’s, warfarin and acenocoumarol.

Physicochemical and Pharmacokinetic Parameters	Warfarin	Acenocoumarol
Molecular weight	308.3	353.3
pKa/LogP	5.0/2.9	5.0/1.98
Vd (L/kg)	0.08–0.12	0.22–0.52
Plasma protein binding (PPB)	>99%	>98%
Plasma concentration (µM/L)	1.5–8	0.03–0.3
Terminal half-life (h)	S-War: 24–33 R-War: 35–58	S-Ac: 1.8 R-Ac: 6.6
Main metabolic pathway	CYP2C9	CYP2C9
Plasma clearance (L/h)	S-War: 0.1–1.0 R-War: 0.07–0.35	S-Ac: 28.5 R-Ac: 1.9

**Table 4 pharmaceuticals-18-00477-t004:** Pharmacogenetic differences in CYP2C9 in the different races integrated in Simcyp^®^ software. In the European Caucasian population, intensive metabolizers (CYP2C9*1) have 73 pmol/mg protein and poor metabolizers (CYP2C9*2, CYP2C9*3)—29 pmol/mg protein in the liver, while for the Asian population, the values are 60 pmol/mg protein and 24 pmol/mg protein, respectively.

CYP2C9 Genotype	European Caucasian	Chinese	Japanese
*1/*1	0.672	0.924	0.96
*1/*2	0.186	0.0024	0
*1/*3	0.111	0.0712	0.0396
*2/*2	0.011	0	0
*2/*3	0.017	0	0
*3/*3	0.003	0.0024	0.0004

**Table 5 pharmaceuticals-18-00477-t005:** PBPK patterns of S-warfarin in a virtual European Caucasian and Asian (Chinese and Japanese) populations.

Population	Cmax (mg/L)	AUC (mg·h/L)
European Caucasian	0.987	17.21
Chinese	1.267	23.58
Japanese	1.205	21.53

**Table 6 pharmaceuticals-18-00477-t006:** Simulations of simultaneous use of S-warfarin with Fluconazole administered in doses of 100, 200, and 400 mg.

Inhibitor of CYP2C9	European Caucasian		Chinese	
	AUCR ± SD	CmaxR ± SD	AUCR ± SD	CmaxR ± SD
Fluconazole 100 mg	1.49 ± 0.16	1.31 ± 0.09	1.06 ± 0.06	1.02 ± 0.01
Fluconazole 200 mg	1.89 ± 0.33	1.56 ± 0.16	1.10 ± 0.11	1.03 ± 0.02
Fluconazole 400 mg	2.51 ± 0.65	1.94 ± 0.29	1.16 ± 0.17	1.04 ± 0.03

**Table 7 pharmaceuticals-18-00477-t007:** Values of CmaxR and AUCR during simultaneous administration of digoxin 0.5 mg with different combinations of verapamil/norverapamil 240 mg.

Digoxin + Verapamil/Norverapamil	CmaxR ± SD	AUCR ± SD
Digoxin + Verapamil/Norverapamil 240 mg p.o. (80 mg/8 h)	1.63 ± 0.28	1.32 ± 0.23
Digoxin + Verapamil/Norverapamil 240 mg i.v. bolus (80 mg/8 h)	1.09 ± 0.05	1.06 ± 0.04
Digoxin + Verapamil (deactivation of P-gp in liver and GIT)/Norverapamil 240 mg	1.25 ± 0.20	1.45 ± 0.24
Digoxin + Verapamil/Norverapamil (deactivation of P-gp in liver and GIT) 240 mg	1.27 ± 0.20	1.51 ± 0.25
Digoxin + Verapamil (deactivation of P-gp in liver)/Norverapamil (deactivation of P-gp in liver) 240 mg	1.25 ± 0.24	1.45 ± 0.20
Digoxin + Verapamil (deactivation of P-gp in GIT)/Norverapamil (deactivation of P-gp in GIT) 240 mg	1.14 ± 0.07	1.07 ± 0.05

**Table 8 pharmaceuticals-18-00477-t008:** CmaxR and AUCR values for dabigatran etexilate (DE) and dabigatran applied with verapamil/norverapamil concomitantly or two hours after DE administration.

	AUCR ± SD	CmaxR ± SD
**Simultaneous use**		
DE + verapamil/norverapamil	2.61 ± 0.75	2.58 ± 0.80
Dabigatran + verapamil/norverapamil	2.34 ± 0.71	2.39 ± 0.75
**Application of the inhibitor after 2 h**		
DE + verapamil/norverapamil	1.55 ± 0.28	1.26 ± 0.32
Dabigatran + verapamil/norverapamil	1.47 ± 0.26	1.42 ± 0.31

## Data Availability

The original contributions presented in this study are included in the article. Further inquiries can be directed to the corresponding author.

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
