# Peer review of "Using Physiologically Based Pharmacokinetic Models for Assessing Pharmacokinetic Drug–Drug Interactions in Patients with Chronic Heart Failure Taking Narrow Therapeutic Window Drugs"

_pharmaceuticals, 2025, doi:10.3390/ph18040477_

Round 1
Reviewer 1 Report
Comments and Suggestions for Authors
The authors propose incorporating pharmacokinetic modeling tools to optimize the treatment of heart failure patients who are polymedicated with drugs that have a narrow therapeutic index. Drug interactions may necessitate the use of a second-line therapy or dose adjustment; in either case, it is crucial for physicians to understand the effects of these interactions. The study retrospectively evaluates data from over 400 patients who meet these criteria.
A computational tool, referred to as “Lexicomp Drug Interaction” but not well-defined by the authors, is used to identify the main interactions among the medications taken by these patients. It would be valuable to include details on the functioning and underlying principles of Lexicomp within the manuscript. The simulations were conducted using data from SimCYP, which I consider the primary limitation of the study. Nevertheless, the authors acknowledge this constraint and have appropriately addressed it in the paper. Ideally, incorporating original data to refine the model before applying it in clinical practice would enhance its reliability.
I recommend revising certain aspects before publication:
- Include an explanation of Lexicomp’s functionality.
- Provide a clearer description of Table 1. There are unexplained question marks and category labels, which I assume originate from the software mentioned in point 1.
- Clarify the header of Table 2 to specify that the values presented are ratios, as this is not mentioned before.
- Revise the text to ensure an impersonal tone, avoiding the use of "we." I have identified instances in lines 109 and 358 where this could be improved.
- Enhance the clarity of Figure 1 by representing the data as a bar chart rather than a three-dimensional pie chart.
- Review Table 4—if the results for "Healthy volunteers" and "European Caucasian" are identical, consider merging them into a single column unless the distinction is relevant. Notably, Table 5 highlights a difference between these two groups, which raises the question: Should they not also be identical in that case?
- Integrate the paragraph following Tables 4 and 5, which discusses enzyme quantities by race, into one of the previous tables for better clarity.
- Strengthen the discussion on dose adjustments by providing concrete examples. Throughout the paper, there is mention of how physicians could use these tools to adjust doses, but supporting this claim with one or more examples would be beneficial.
- Correct the abbreviation list, as "P-gp" appears twice.
These revisions would significantly improve the clarity and impact of the manuscript.
Author Response
Dear reviewer,
Thank you for your comments and suggestions for improving the article. These are our responses to the issues raised.
Comments 1: Include an explanation of Lexicomp’s functionality.
Response 1: We've added the core functionality of Lexicomp.
Comments 2: Provide a clearer description of Table 1. There are unexplained question marks and category labels, which I assume originate from the software mentioned in point 1.
Response 2: Since we have added the information on how the software works, we feel that there is no need for explanations in the table.
Comments 3: Clarify the header of Table 2 to specify that the values presented are ratios, as this is not mentioned before.
Response 3: We have added the information for clarity.
Comments 4: Revise the text to ensure an impersonal tone, avoiding the use of "we." I have identified instances in lines 109 and 358 where this could be improved.
Response 4: The sentences have been paraphrased.
Comments 5: Enhance the clarity of Figure 1 by representing the data as a bar chart rather than a three-dimensional pie chart.
Response 5: We intend to keep Figure 1 as we have presented it. Presenting it this way is automatically generated by the program, and we think it is visual enough to illustrate what is at stake.
Comments 6: Review Table 4—if the results for "Healthy volunteers" and "European Caucasian" are identical, consider merging them into a single column unless the distinction is relevant. Notably, Table 5 highlights a difference between these two groups, which raises the question: Should they not also be identical in that case?
Response 6: We thought it would be better to remove the healthy volunteers from Tables 4 and 5 to avoid confusion. The main purpose is to compare the European race with the Asian race.
Comments 7: Integrate the paragraph following Tables 4 and 5, which discusses enzyme quantities by race, into one of the previous tables for better clarity.
Response 7: We have added information about enzyme quantities in different races in the heading of Table 4
Comments 8: Strengthen the discussion on dose adjustments by providing concrete examples. Throughout the paper, there is mention of how physicians could use these tools to adjust doses, but supporting this claim with one or more examples would be beneficial.
Response 8: Thanks for the suggestion, but adding a specific example with the recalculation of doses for the specific case would shift the focus of the paper. We have not set this as a future goal by chance. We feel that the examples discussed are sufficient to illustrate the capabilities of such a simulator.
Comments 9: Correct the abbreviation list, as "P-gp" appears twice.
Response 9: The mistake has been corrected.
Reviewer 2 Report
Comments and Suggestions for Authors
The authors underscore the significance of PBPK models in elucidating the pharmacokinetic aspects of drug-drug interactions and devising dosage regimens for patients with chronic heart failure. These models are essential for drugs with a narrow therapeutic window. This research further offered recommendations for choosing an alternative drug delivery route to mitigate drug-drug interactions, thereby reducing the risk of sub-therapeutic and toxic levels. Please address the following comments for further consideration of the manuscript.
- The authors utilized Lexicomp software to predict potential drug-drug interactions. Please provide a detailed explanation of the input factors necessary to evaluate interaction risk and elucidate how the software assigns risk ratings to interactions, ranging from minor to major, and offers recommendations for managing or avoiding these interactions.
- The authors compared the simulated Cmax and AUC results with clinical outcomes. It is crucial to provide a comprehensive explanation of the model's design and the assumptions underlying the studied conditions for CYP3A4, CYP2C9, and P-GP interactions. Additionally, a detailed discussion of the pharmacokinetic parameters affecting Cmax and AUC, as well as bioavailability, is necessary.
- Figure 1 lacks clarity and requires additional explanation. Please provide definitions for HLM, fm, and fe. Furthermore, there is insufficient discussion on the alternative pathways, their quantification, and the reasons for the elevated Simvastatin Cmax and AUC.
- The conclusions section could be improved. Rcommended to provide a comprehensive summary of the model-calculated parameters and clinical outcomes, emphasizing how the model accurately predicted drug-drug interactions and the Cmax and AUC values.
Author Response
Dear reviewer,
Thank you for your comments and suggestions for improving the article. These are our responses to the issues raised.
Comments 1: The authors utilized Lexicomp software to predict potential drug-drug interactions. Please provide a detailed explanation of the input factors necessary to evaluate interaction risk and elucidate how the software assigns risk ratings to interactions, ranging from minor to major, and offers recommendations for managing or avoiding these interactions.
Response 1: We've added the basic information about the Lexicomp software, and we think that's enough. More detailed information can be found on the manufacturer's website. We do not feel that all the details are necessary for the purposes of our article.
Comments 2: The authors compared the simulated Cmax and AUC results with clinical outcomes. It is crucial to provide a comprehensive explanation of the model's design and the assumptions underlying the studied conditions for CYP3A4, CYP2C9, and P-GP interactions. Additionally, a detailed discussion of the pharmacokinetic parameters affecting Cmax and AUC, as well as bioavailability, is necessary.
Response 2: We have provided a comprehensive explanation of the model's design and assumptions in Materials and Methods, where the software is described. Information on parameters affecting Cmax, AUC, and bioavailability is given briefly in the section Discussion.
Comments 3: Figure 1 lacks clarity and requires additional explanation. Please provide definitions for HLM, fm, and fe. Furthermore, there is insufficient discussion on the alternative pathways, their quantification, and the reasons for the elevated Simvastatin Cmax and AUC.
Response 3: Definitions were given for HLM, fm, and fe, and a comment was added about the observed effects on Cmax and AUC.
Comments 4: The conclusions section could be improved. Recommended to provide a comprehensive summary of the model-calculated parameters and clinical outcomes, emphasizing how the model accurately predicted drug-drug interactions and the Cmax and AUC values.
Response 4: We have added an addendum to the conclusion, and it sounds better this way now.
Round 2
Reviewer 2 Report
Comments and Suggestions for Authors
My comments were addressed. No further comments from me.